# An-m-Health Intervention Using Smartphone App to Improve Physical Activity in College Students: A Randomized Controlled Trial

**DOI:** 10.3390/ijerph19127228

**Published:** 2022-06-13

**Authors:** Hala K. Al-Nawaiseh, William A. McIntosh, Lisako J. McKyer

**Affiliations:** 1Department of Nutrition and Food Technology, Faculty of Agriculture, The University of Jordan, Amman 11942, Jordan; 2Department of Nutrition and Food Science, Faculty of Agriculture and Life Science, Texas A&M University, College Station, TX 5429-3011, USA; w-mcintosh@tamu.edu; 3Department of Health Promotion and Community Health Sciences, School of Public Health, Texas A&M University, College Station, TX 5429-3011, USA; lisako.mckyer@tamu.edu

**Keywords:** m-Health, physical activity, body mass index, college students

## Abstract

Using m-Health apps can provide researchers and others with an effective way for improving physical activity (PA) and healthy lifestyle behaviors. The promotion of health should move from a model focused on the physical and biological basis of illness and towards a focus on the behavioral changes that support health. Therefore, the aims of the current study were to improve PA (step-counts) and body weight using a theory-based m-Health app. A 12-week randomized treatment trial was carried out at Texas A&M University, Texas, college station. College students (*n* = 130) were recruited. They were randomized in an equal ratio of 1:1 to intervention (m-Health app) (*n* = 65) and control (*n* = 65) conditions. The response rate was (87.6%). Both groups utilized a Smartphone app. The intervention group received PA goals of (10,000 steps/day), using an m-Health app. The control group was provided with information related to daily recommended PA levels. The primary change was daily step count between the baseline and follow-up. The secondary outcome was the body mass index (BMI). Descriptive statistics were used to summarize the baseline differences between the control and intervention groups. Independent sample *t*-test were used for comparison between the intervention and control groups. Post-intervention PAs were higher for the intervention group (mean = 54,896.) vs. control group (mean = 45,530.12; *p* < 0.05). The intervention group’s step-counts increased significantly (pre-mean = 40,320.38 steps per week; post-mean = 54,896.27 steps per week, *p* < 0.05). The body-weight changes were significant among the intervention group (*p* < 0.05). m-Health apps can increase PA and improve body weight, with goal setting and feedback as key intervention components. Future studies should personalize PA goals and feedback.

## 1. Introduction

College students are a group that has a rate of 52.2% of physical inactivity, and 34.4% of them are overweight or potentially vulnerable to becoming overweight/obese later life [1,2]. Therefore, emphasizing the importance of PA among college students is an important effort to prevent serious health problems, such as obesity during and after college [1]. It is an exciting time for PA research, given the evolving technology for intervening with PA, a new focus on novel behavioral targets (e.g., sedentary behavior), and increased attention on scaling evidence-based interventions for maximal public health impacts [3].

### m-Health Application as a Support for Improving Physical Activity

Growing evidence has been established about changing behavior using m-Health tailoring applications (apps) and their effective role in changing lifestyle risk factors, such as PA [4,5]. Therefore, theoretically based behavior changes interventions that use m-Health apps could provide researchers and others with an effective way of improving PA and healthy lifestyle behaviors. Unfortunately, interventions using m-Health apps have dealt with mainly women and for limited times [6,7]. An emphasis on short-term behavioral change present in most efficacy research provides little insight on what is needed to maintain or institutionalize an intervention [8].

It has been found that longer intervention could be more appropriate to determine if the target behavior compliance is maintained for a greater time [8]. To address this gap, the current study involved both male and female college students in a relatively longer intervention period (12-weeks) than other studies. There is little attention to measuring the effect of an m-Health app intervention program designed to improve the PA of college students, even though several studies highlighted an m-Health app intervention as a main channel that demonstrated its effectiveness in health-related changes. Despite the potential of mobile applications and growing public and researcher interest in their use, there are few randomized control trials (RCTs) of m-Health apps as a healthy lifestyle intervention that focuses on the education and self-monitoring of PA to our knowledge. As a result, no such interventions have been conducted with healthy college students [7,9]. The goal of this 12-week RCT is to promote healthy lifestyle habits connected with PA (step counts) and body weight among college students using the m-Health app (aged 18–30). We hypothesized that those using an m-Health lifestyle promoting app will increase PA (step counts) over a 12-week period when compared with control group. We hypothesized that encouraging students to engage with features of an app (e.g., goal-setting and self-monitoring) will significantly increase their step counts compared to giving standard physical activity recommendations.

## 2. Materials and Methods

### 2.1. Design and Setting

The study design was a 12-week-RCT, recruiting 130 students from Texas A&M University/College Station campus to improve healthy lifestyle behavior associated with PA using a theory-based m-Health app intervention among college students. Various recruitment strategies were used (email, advertisements posted around the university, and classroom recruitment) from January 2018 to March 2018, and all outcome assessments were completed by 1 July 2018. Following the screening visit, participants who were interested in the study were screened for pre-specified inclusion and exclusion criteria. Eligibility criteria included apparently healthy college students aged 18–30 years, interested in a healthy lifestyle behavior, with the ability to commit the necessary time and effort to be participants in the study, with a body mass index (BMI): BMI ≥ 18.5, capable of performing walking PA, and that were owners of a Smartphone (i-phone or Android). However, the exclusion criteria were participants who had no Smartphone or were in another program for weight reduction, were pregnant, lactated, had bariatric or recent surgery, had any diagnosed chronic diseases such as musculoskeletal disorders, heart failure, diabetes mellitus, hypertension, dyslipidemia, or cancer, they could not undertake moderate exercise for any reason, and/or had already used the (Pacer) pedometer app.

This study was conducted according to the Declaration of Helsinki, and all procedures involving research study participants were approved by the Institutional Review Board (IRB) at Texas A&M University and the human research ethics committee (IRB2 # 018-0022D). Informed consent was obtained from all study participants. Upon completing the baseline assessment, eligible participants were randomly assigned to the intervention group (m-Health app) (*n* = 65) or the control group (*n* = 65) in an equal ratio of 1:1. The randomization process was performed using the “Research Randomizer ORG” computer software program (www.randomizer.org/form.htm) (access 18 January 2018), and the researcher was responsible for generating the allocation sequence. The researcher cracked the randomization code at the end of week one. Participants were aware that two groups existed, but they were blinded to the nature of each group.

### 2.2. Sample Size Calculation

The sample size for the RCT was calculated based on previous RCT studies and calculated after considering several assumptions [10,11]; most RCTs have used two-sided tests to detect between-group differences. It would be possible to detect a relative increase of at least 20% in the change of the average step count between the intervention and control groups, with 80% power at a significance level of the (α = 0.05) threshold for statistical significance and a medium effect size of Cohen’s (d = 0.5 one-half standard deviation) [12], and there would be a drop-out rate of 15%. The estimated sample size was 130 college students, who were randomized to receive either the intervention or control groups (65 participants per group were required) (Figure 1).

### 2.3. Need Assessment Phase: Phase One

#### 2.3.1. Anthropometric Measurements

At the baseline (one week prior to the intervention), anthropometric measurements, including weight, height, BMI, and percentage of body fat, were obtained. Weight, BMI, and percentage of body fat were determined using the TANITA Body Composition Analyzer (SC331S) [13]. According to guidelines stated by the National Institutes of Health, weight status was classified into four categories: underweight (BMI ≤ 18.5), normal weight (BMI between 18.5–24.9), overweight (BMI between 25–29.9), and obese (BMI ≥ 30) [13]. All the participants had the Smartphone app (pacer) installed on their phones to track their daily step counts to provide a baseline measurement of their PA levels. The program featured automatic step count feedback and tracking, as well as a visually pleasing display of step count history and goal completion. Each participant had a 20-step test to calibrate the phone’s sensitivity level for different persons, ensuring that their step counts were captured accurately [10]. All participants were requested to carry their Smartphone during walking hours for the week following their screening visit and to maintain their typical PA levels. At the end of this period, each participant met with a researcher, who received the previous week’s step-count data. A 20-step test was carried out with each participant to calibrate the sensitivity level of the telephone for different individuals to ensure that their step counts were recorded accurately.

The initial face-to-face (FTF) meeting with all study participants (45 min) served as an orientation to the project. Each group of five people received a presentation about a healthy lifestyle and the clinical benefits of weekly recommendations of PA.

#### 2.3.2. Measures

In phase one, online surveys administered at the baseline included questions addressing the following demographic data: age, gender, ethnicity/and race. Physical activity was assessed using a pedometer-based m-Health app (steps/week). Weight status was objectively measured by height, weight, BMI, and body fat percentage.

### 2.4. m-Health Applications Intervention: Phase Two

#### 2.4.1. Physical Activity: Pacer Pedometer App

##### Commercially Available App

We used one of the most popular publicly available Smartphone apps for improving the PA (Step counts) (Pacer). It has goal setting functionality, self-monitoring of step counts, calories expended, and automatic performance feedback through the graphic display of step-count history.

##### Intervention Group

The intervention group received PA goals in terms of 10,000 steps/day and was informed that this value is roughly equivalent to 30 min of walking per day (along with their normal activity). They received information about the benefits of exercise and instructions on how to use the app. The researcher also demonstrated the usability features of the mobile phone app to the intervention group (using standardized instructions) and encouraged this group to use the app to monitor their steps and obtain feedback, in order to achieve their target goals. They were instructed also to keep their phones charged and to always carry it during waking hours. By the end of each week (week 2 to week 12), each participant in the intervention group was contacted via SMS/e-mail and asked to share their step count data with the researcher.

##### Control Group

Participants in the control group were provided with information related to daily recommended PA levels (i.e., 30 min daily) and information highlighting the benefits of walking regularly, without being observed or requiring interaction with the researcher. Members of the control group did not use the pacer app beyond the first week of the assessment phase (Baseline Assessment) until week 11, when they were contacted for a week 12 follow-up assessment, and they received no extra intervention.

### 2.5. Evaluation of m-Health Applications Intervention: Phase Three

#### Measures

Physical Activity: (Pacer) pedometer m-Health based-app: Feedback and tracking of step counts, calories expended, and step-count history. It was assessed across the 4 time points. Time was coded on a continuous scale, with 0 at the baseline, and 4, 8, and 12-week follow-up assessments. However, for the present analysis, we compared only the baseline and week 12 of intervention.Weight Status: Objectively measured height, weight, BMI, and body fat percentage was obtained at the end of 12 weeks of intervention.

### 2.6. Data Analysis

All analyses were conducted using IBM SPSS Statistics for Windows, Version 24.0. Armonk, NY: IBM Corp with the 2-sided level of α level set at *p* < 0.05; data were assessed for normality. The variables were expressed as percentage for categorical variables and means plus standard deviation for continuous data. Descriptive statistics were used to summarize the baseline differences in participant characteristics between the intervention and control groups. Paired *t*-tests were used for comparison between pre- and post-intervention variables (pedometer measures, and anthropometric) for both intervention and control groups. Independent sample *t*-tests were used for comparison between the intervention and control groups.

## 3. Results

### 3.1. Demographics

Demographic characteristics of participants were comparable at the baseline across intervention and control groups (Table 1). The two groups combined had a mean age of 21.12 (±2) years, a mean BMI of 22.87 (±3.8), and a mean fat percentage of 23.10 (±9.9). Participants were predominantly female (92 (80.7%)) and white (56 (49.1%)). At the baseline, there were no statistically significant differences between the two groups in terms of these characteristics (*p* > 0.05, Table 1).

College student participants across both the control and intervention groups averaged 43,342.59 (±16,584.56) steps/week, and the difference in steps/week between the intervention and control groups at the baseline were not statistically significant (control: 46,260.60 vs. intervention: 40,320.37, *p* = 0.056). Of the 130 participants, 114 (87.6%) completed the follow-up; the most loss to the follow-up occurred during the follow-up week (week # 12).

### 3.2. Primary Outcomes

#### Physical Activity (PA) (Step Counts/Week)

A *t*-test revealed a significant increase in weekly step counts from the baseline (week 1) to the follow-up (week 12) among the intervention group (mean differences = −14,575.89, t55 = −6.113, *p* = 0.00). However, there was no significant difference between the baseline (week 1) and follow-up (week 12) step counts among the control group (mean differences = 730.48, t57 = 1.72, *p* = 0.09; Table 2). A *t*-test of the differences in step counts at the follow-up (week 12) revealed that the participants in the intervention group had a significantly higher increase in step counts (mean = 54,896.27) than those in the control group (mean = 45,530.12) (t112 = 3.255, *p* = 0.00; Table 2).

### 3.3. Secondary Outcomes

#### Anthropometric Measurements

Changes for the secondary outcomes (body weight, fat%, and BMI) among both the intervention and control groups are reported in Table 3. No changes were found for the secondary outcomes of body weight, fat%, and BMI among the control group. The changes were as follows: body weight (mean ± SD: −0.08 ± 1.57 kg; *p* = 0.71), fat% (mean = 0.32 ± 1.49%; *p* = 0.11), and BMI (mean = −0.05 kg/m^2^ ± 0.62; *p* = 0.54), among the control group. However, the intervention group significantly reduced their body weight (mean ± SD: 0.42 ± 1.23 kg, *p* < 0.05), but there were no significant changes for fat% (mean = −0.87 ± 4.88%; *p* = 0.19) or for BMI (mean = 0.05 ± 0.80; *p* = 0.64).

Table 4 indicates that no significant differences were found for the post-intervention secondary outcomes of body weight, fat%, and BMI between the control and intervention groups. The changes were as follows: body weight (mean = 2.95 kg; *p* = 0.23), fat% (mean = −1.28%; *p* = 0.46), and BMI (mean = 1.19 kg/m^2^; *p* = 0.10).

## 4. Discussion

The present study provided valuable insights into the use of m-Health apps to investigate theory-based behavioral interventions to improve healthy lifestyle behaviors among college students, which may be suitable for widespread dissemination and implementation. Findings from the current RCT indicate that a 12-week m-Health intervention using an m-Health app resulted in a significant improvement in PA (step counts) and body weight in college students over and above the provision basic information on recommendations among young adults. Substantial outreach efforts were needed to recruit the participants, but eligibility rates were high among those who were screened, and the retention through 12 weeks was excellent (88%).

The use of a pedometer-based m-Health app was found to increase the PA over a 12-week period, when compared with data from the control group. The intervention group achieved a significant increase of over 14,575 steps per week (an increase of approximately 36% in activity levels), equivalent to about seven miles/week. A comparable step count increase (1029 steps/day) over a period of 8 weeks had been reported in the SMART MOVE trial among the intervention group, who received PA goals of 10,000 steps/day [10]. Moreover, comparable step counts that increased over a mean period of 18 weeks of intervention were associated with a significant reduction in BMI [14].

Our data are comparable to the data from [14], who reported that a pedometer-based m-Health intervention increased the users PA (step counts) by an average of 26.9% over the baseline PA. Furthermore, it was found that, during a mean of 6 years, a 2000 step count/day increase was associated with a 10% relative reduction of cardiovascular diseases (CVD) [15]. The high effect size (ES) (ES = 0.82) of the current study and increase in step counts were comparable to the results found in a meta-analysis, which found that the interventions with 10,000 steps/day as a target goal had the highest effect size (ES = 0.84, 95%CI = 0.43, 1.24) [16]. This evidence will help us in designing an optimal m-Health intervention that optimizes the improvement of PA engagement. This substantial change is clinically meaningful and, if continued, is expected to result in numerous health benefits, including a decreased risk of obesity, type 2 diabetes mellitus (T2DM), and CVDs [17].

The current study followed the intervention group throughout the intervention period weekly and reported an incremental increase in step counts on a month-by-month basis. It has been proposed that sharing PA (step counts) on a regular basis may act as an intrinsic motivational factor. It was also reported that one limitation of previous research was measuring step counts pre and post intervention [18].

Notably, in the current study, the baseline PA levels for both control and intervention groups were low (approximately 5600 steps/day), which was less than the 7000–8000 steps/day, a reasonable threshold of free-living PA that is also associated with the current public health guidelines based on minimal amounts of time spent in Moderate, Vigorous Physical Activity (MVPA) [19].

By the end of the intervention, the intervention group progressed from being classified as “low active” (5000–7499 steps/day) to classified as “somewhat active” (7500–9999 steps/day); they had reached the minimum guideline targets (7842 steps/day) suggesting that pedometer-based m-Health apps are an effective method of promoting PA among college students and allow them to meet suggested public health recommendations, where it is important to develop healthy behaviors.

The main concept of using pedometer m-Health apps to improve PA is that the “real time” step count feedback increases an individual’s consciousness about how personal behavioral choices affect their PA. Although the weight loss was not a target of the increase in daily step counts, the intervention group significantly decreased their body weight (*p* < 0.05). However, intervention participants’ health outcomes (fat percentage and BMI) did not change significantly (t55 = −1.33, *p* > 0.05 and t55 = 0.47, *p* > 0.05; respectively). m-Health app interventions primarily promoted the healthy lifestyle behaviors that may eventually result in a healthier BMI.

Our findings are consistent with [8] findings who reported a significant weight reduction pre- to post-intervention among the intervention group (−1.6 (2.6) kg), as a result of a 12-week period pedometer based on an m-Health intervention with a recommendation of 45–60 min/day of moderate intensity PA. Findings from the current study suggest that pedometer-based m-Health app intervention that targets college students may facilitate their engagement in PA over a relatively short period of time (12 weeks).

### 4.1. Limitations

Limitations of the present study include a relatively small sample size; there was a large proportion of females in the sample, which may limit the generalizability of the findings, and the mean BMI was within normal body weight range. In addition, pedometer-based m-Health apps cannot evaluate the intensity and speed of activity. Finally, the study was of relatively short duration (3-months) with no extended follow-up. A longer intervention period would have provided clearer insight into the maintenance of behavioral changes.

### 4.2. Strength

The present study found an effective m-Health app intervention among targeted college students. In addition, the study included the randomized controlled design powered to detect the effectiveness of using an m-Health application to improve the healthy lifestyles among college students. There was a high retention rate at the follow-up (88%) and, finally, a limited number of exclusion criteria, therefore strengthening external validity and helping to facilitate the implementation [9].

### 4.3. Implications for Research and Practice

The promotion of health should move from a model focused on the physical and biological basis of illness and towards a focus on the behavioral changes that support the health. These changes seem to be effectively facilitated through m-Health app intervention programs.

The current study demonstrated that a 12-week RCT m-Health intervention program was an effective way to significantly improve PA (step counts), body weight, and other healthy lifestyle benefits in college students. Although, weight management is not a behavior, but rather it is an outcome of behavioral changes. Pedometer-based m-Health apps facilitate the behavioral change techniques (BCTs) of goal setting, self-monitoring, and action planning through automatic visual feedback and consequences for PA (step counts).

Significant and clinically meaningful improvements in PA and healthy lifestyle behaviors are harder to achieve during a relatively short period of intervention (12 weeks). Therefore, a larger RCT with a larger population group would allow the generalizability of the current results. In addition, longer follow-up interventions are required to allow positive physiological changes, such as high blood pressure, to occur and to examine the long-term sustainability of such improvement.

## 5. Conclusions

Promotion of healthy lifestyle behaviors among college students is a priority in line with the WHO strategies to improve health through engaging in PA, in order to reduce the risk of diseases burden in later life. The current study demonstrated that a 12-week RCT m-Health intervention program was an effective way to significantly improve PA (step counts). Although we found a significant improvement in PA (step counts) by setting a goal of 10,000 steps/day among all participants, future research should be moving toward the personalization of m-Health intervention based on an individual’s unique profile, which seems to be more effective and increases the likelihood of successful behavior changes, and it leads to greater engagement in m-Health intervention. It would be of interest for future research to include other objective measures of fitness, such as maximal oxygen uptake and heart rate. In conclusion, m-Health is an emerging field of health interventions and research that offers an alternative, usually effective method of improving physical activity among college students. It represents a unique opportunity in public health to explore, understand, and effectively improve PA.

## Figures and Tables

**Figure 1 ijerph-19-07228-f001:**
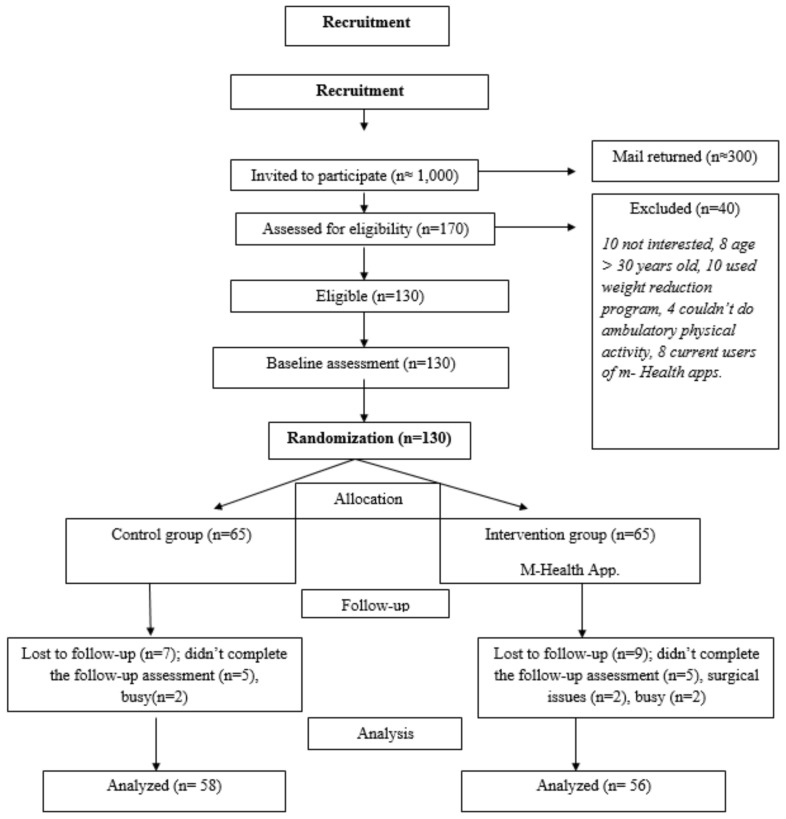
Participant flow diagram.

**Table 1 ijerph-19-07228-t001:** Baseline characteristics of the participants according to the group allocation.

	Overall	Control	Intervention	*p*-Value
**Total, no.**	114	58	56	
**Age (years), mean (SD)**	21.12 (2.2)	21.46 (2.6)	20.76 (1.8)	0.94
**Female, no. (%)**	92 (80.7)	46 (79.3)	46 (82.1)	0.000
**Male, no. (%)**	22 (19.3)	12 (20.7)	10 (17.9)	
**Race category, no (%):**				
White	56 (49.1)	23 (39.7)	33 (58.9)	0.000
Black or African American.	12 (10.5)	7(12.1)	5 (8.9)	
American Indian or Alaska Native.	16 (14.0)	11(19.0)	5 (8.9)	
Asian	20 (17.5)	10 (17.2)	10 (17.9)	
Other	10 (8.8)	7(12.1)	3 (5.4)	
**Weight, mean (SD)**	66.19 (12.8)	67.39 (13.8)	64.94 (11.6)	0.307
**Fat percentage, mean (SD)**	23.10 (9.9)	23.1 (9.4)	23.15 (10.5)	0.961
**BMI, mean (SD)**	22.87 (3.8)	23.42 (4.4)	22.32 (2.9)	0.124
**Baseline step count, mean (SD)**	43,342.59 (16,584.56)	46,260.60 (16,261.52)	40,320.37 (16,515.63)	0.056

Differences between the groups at the baseline were evaluated with Independent-Sample *t*-Test, and Chi-Square, *p*-value < 0.05.

**Table 2 ijerph-19-07228-t002:** Changes in physical activity (PA) (step counts/week) among intervention and control groups.

Step Counts		N	Mean (SD)	Mean Differences	t	df	*p-*Value
**Intervention**	Pre-intervention	56	40,320.38 (16,515.63)	−14,575.89	−6.113	55	0.00
	Post-intervention	56	54,896.27 (14,992.35)				
**Control**	Pre-intervention	58	46,260.60 (16,261.52)	730.48	1.726	57	0.90
	Post-intervention	58	45,530.12 (15,703.41)				
**Post-intervention**	Control	58	45,530.12 (15,703.41)	−9366.15	−3.255	112	0.00
	Intervention	56	54,896.27 (14,992.35)				

The mean difference is significant at the 0.05 level.

**Table 3 ijerph-19-07228-t003:** Anthropometric variables for the intervention and control group.

	Variables		N	Mean (SD)	Mean Differences (SD)	t	df	*p-*Value
**Intervention**	**Body weight** **(kg)**	Pre-intervention	56	64.94 (11.6)	0.42 (1.23)	2.56	55	0.01
		Post-intervention	56	64.52 (11.6)				
	**Fat (%)**	Pre-intervention	56	23.15 (10.5)	−0.87 (4.88)	−1.33	55	0.19
		Post-intervention	56	24.01 (9.2)				
	**BMI (kg/m^2^)**	Pre-intervention	56	22.32 (2.9)	0.05 (0.80)	0.47	55	0.64
		Post-intervention	56	22.27 (2.9)				
**Control**	**Body weight (kg)**	Pre-intervention	58	67.39 (13.83)	−0.08 (1.57)	−0.37	57	0.71
		Post-intervention	58	67.47 (14.42)				
	**Fat (%)**	Pre-intervention	58	23.06 (9.36)	0.32 (1.49)	1.63	57	0.11
		Post-intervention	58	22.74 (9.13)				
	**BMI (kg/m^2^)**	Pre-intervention	58	23.41 (4.4)	−0.05 (0.62)	−0.61	57	0.54
		Post-intervention	58	23.46 (4.6)				

The mean difference is significant at the 0.05 level.

**Table 4 ijerph-19-07228-t004:** Post intervention comparisons of anthropometric variables between the control and intervention groups.

Post-Intervention	Group	N	Mean (SD)	Mean Differences	t	df	*p*-Value
**Body weight (kg)**	Control	58	67.47 (14.42)	2.95	1.201	112	0.23
	Intervention	56	64.52 (11.6)				
**Fat (%)**	Control	58	22.74 (9.13)	−1.28	−0.744	112	0.46
	Intervention	56	24.01 (9.21)				
**BMI (kg/m^2^)**	Control	58	23.46 (4.6)	1.19	1.643	112	0.10
	Intervention	56	22.27 (2.9)				

The mean difference is significant at the 0.05 level.

## Data Availability

All data generated or analyzed during this study are included in this published article. The datasets used and/or analyzed during the current study are available from the corresponding author on reasonable request. **Clinical Registry**: ISRCTN16108619 https://doi.org/10.1186/ISRCTN16108619.

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
