# Peer review of "An-m-Health Intervention Using Smartphone App to Improve Physical Activity in College Students: A Randomized Controlled Trial"

_ijerph, 2022, doi:10.3390/ijerph19127228_

Round 1

Reviewer 1 Report

Dear editors, thanks for the opportunity to do this review. This is an interesting article that describes helpful research about the use of mHealth to improve physical activity among students. Albeit the topic is not new, I believe that this research can be helpful for actual clinical practice and other researchers. Nevertheless, I think that there are some minor aspects that perhaps could be revised. These are the following:

In Materials, the authors write that «The study design was a 12-week-RCT, recruiting 130 students from Texas A&M University/ College Station campus to improve healthy lifestyle behavior associated with PA using a theory-based m-Health app intervention among college students. Various recruitment strategies were used (email, advertisements posted around the university, and classroom recruitment)». The main problem of this study is the potential selection bias. Thus, the authors should assess it, so the readers can better understand the external validity of their conclusions.

In this sense, the authors describe that «Eligibility criteria included apparently healthy college students aged 18-30 years, interested in a healthy lifestyle behavior, and have ability to commit the necessary time and effort to participants in the study». Thus, the sample is clearly biased and I would kindly invite the authors to assess it. For example, 80.7% of the sample were women, who usually are more concerned with health, and who usually use more digital technologies for health, than men.

Albeit the authors describe how they performed the sample size estimation, the sample perhaps seems a bit low. I would invite them to make a better assessment of the sample size, if possible. This is just a proposal.

The authors describe that «We used one of the most popular publicly available Smartphone apps for improving the PA (Step counts) (Pacer)». It would be interesting how they stated that this app was one of the most popular apps. Was it free? (I understand that yes) Why did they not choose the most popular? Or is it the most popular? Who defines what app is the most popular? Perhaps a reference would be useful.

In the limitations section, I would invite the authors to assess better the potential selection bias, and the sample size, as they are important to understand the external validity of their conclusions.

I would suggest using a specific section for the conclusions.

Numbers like 43342.59 are difficult to read. Perhaps using the format 43,342.59 would be easier.

Please define PA at the beginning of the Abstract.

Author Response

  1. In Materials, the authors write that «The study design was a 12-week-RCT, recruiting 130 students from Texas A&M University/ College Station campus to improve healthy lifestyle behavior associated with PA using a theory-based m-Health app intervention among college students. Various recruitment strategies were used (email, advertisements posted around the university, and classroom recruitment)». The main problem of this study is the potential selection bias. Thus, the authors should assess it, so the readers can better understand the external validity of their conclusions.

 In this sense, the authors describe that «Eligibility criteria included apparently healthy college students aged 18-30 years, interested in a healthy lifestyle behavior, and have ability to commit the necessary time and effort to participants in the study». Thus, the sample is clearly biased and I would kindly invite the authors to assess it. For example, 80.7% of the sample were women, who usually are more concerned with health, and who usually use more digital technologies for health, than men.

We completely agree with the reviewer’s assessment regarding the selection bias. However, random emails had been sent through the university networks, as well as random posting for advertisement through the university campus to obtain the representative sample. Therefore, this way of randomization in sample selection enhanced the external validity. However, we addressed through the limitation section that small sample size; and large proportions of female in the sample, limit the generalizability of the current findings. Moreover, we addressed that a high retention rate at follow-up (88%) and, limited number of exclusion criteria strengthening external validity and help to facilitate the implementation.

  1. Albeit the authors describe how they performed the sample size estimation, the sample perhaps seems a bit low. I would invite them to make a better assessment of the sample size, if possible. This is just a proposal.

  • We agree with the reviewer that the sample size of the current study is low. However, it is comparable to the sample size in previous Randomized Controlled Trial and calculated based on different assumptions that had been explained in Manuscript.

  1. The authors describe that «We used one of the most popular publicly available Smartphone apps for improving the PA (Step counts) (Pacer). It would be interesting how they stated that this app was one of the most popular apps. Was it free? (I understand that yes) Why did they not choose the most popular? Or is it the most popular? Who defines what app is the most popular? Perhaps a reference would be useful.

  • We considered it is one of the most popular since it has 4.8/5 star rating in the App Store, and several fitness websites indicated that Pacer app is considered the most popular since it is free and suitable to download for both iOS and Android.

  • Several studies indicated that pacer has a high accuracy levels regarding to step counts. Pacer, Runtastic, and Argus Android apps were more valid than similar iOS apps in all conditions. However, both iOS and Android apps showed high accuracy levels regarding step count with no systematic errorsReference: Criterion Validity of iOS and Android Applications to Measure Steps and Distance  in Adults (2021)

  1. In the limitations section, I would invite the authors to assess better the potential selection bias, and the sample size, as they are important to understand the external validity of their conclusions.

We completely agree with the reviewer’s assessment regarding the potential selection bias. However, as we addressed in point 1, that we depended on Randomization for participants selections, and we addressed on limitation section that small sample size; and large proportions of female in the sample, limit the generalizability of the current findings, as well as affects on the external validity.  Moreover, a high retention rate at follow-up (88%), and limited number of exclusion criteria strengthening external validity

  1. I would suggest using a specific section for the conclusions.

We agree with the reviewer’s assessment regarding the need for specific section for the Conclusion. We add this part to the Manuscript (highlight):

          Conclusion

           “Promotion of healthy lifestyle behaviors among college students is a priority in line with the WHO strategies to improve health through engaging in PA, in order to reduce the risk of diseases burden in later life. The current study demonstrates that a 12-week RCT m-Health intervention program was an effective way to significantly improve PA (step counts). Although we found a significant improvement in PA (step counts) by setting goal of 10,000 steps/day among all participants, future researches should be moving toward personalization of m-Health intervention based on an individual’s unique profile, which seems to be more effective and increases the likelihood of successful behavior changes, and leads to greater engagement in m- health intervention. It would be of interest for future research to include other objective measures of fitness such as maximal oxygen uptake and heart rate. In conclusion, m-Health is an emerging field of health interventions and research, which offers an alternative, usually effective methods of improving physical activity among college students. It represents a unique opportunity in public health to explore, understand, and effectively improve the PA.

  1. Numbers like 43342.59 are difficult to read. Perhaps using the format 43,342.59 would be easier.

We agree with the reviewer’s assessment regarding to this point, and we modified the formatting through the manuscript (Highlight)  

  1. Please define PA at the beginning of the Abstract:

The term Physical Activity (PA) is added to abstract (Highlight). 

Reviewer 2 Report

Dear authors,

This article nicely describes a M-health RCT intervention, well designed and completed.

However, the manuscript needs a(nother) language revision, as the grammar is often poor, and occasionally with unnecessary repetitions.  

Please see my comments below:

Page

Line

Text citation

Comment

1

12

……”m-Health”.

Please write the full wording and the abbreviation in a parenthesis the first time an abbreviation is mentioned.

1

27

….” Mean = 54896.”

Please write “steps per week” (if that is correct) after the step-counts. And please add in all step counts throughout the article.

2

46-47

“Mobile health….”

Please  full wording only first time mentioned.

2

55 - 58

….” With several studies highlighted an m-Health app intervention as a main channel in which included interventions that demonstrated its effectiveness in health-related changes, there is little attention to measuring the effect of an m-Health app intervention program designed to improve PA of college students..”

Please rephrase /correct grammar.

 Hard to understand the meaning.

2

59-62

“Despite the potential mobile apps and growing interest in their utilization among the public and researchers, to the best of our knowledge, relatively few randomized control trials (RCTs) of m-Health apps as a healthy lifestyle intervention in itself that focuses on education and self-monitoring of PA. Thus, no such interventions have involved populations such as healthy college students [11- 12]. Therefore, the purpose of this a 12 week RCT is….”

Please correct grammar / add verb / rephrase

2

78-88

….” Following the screening visit, participants interested in the study were screened for pre-specified inclusion and exclusion criteria. Eligibility criteria included apparently healthy college students aged 18-30 years, interested in a healthy lifestyle behavior , and have ability to commit the necessary time and effort to participants in the study, having a body mass index (BMI): BMI≥ 18.5, capable of performing walking PA , and owned a Smartphone (i-phone or Android). However, the exclusion criteria were participants who had no Smartphone or were in another program for weight reduction, pregnancy, lactation, had bariatric or recent surgery, or had any of the diagnosed chronic diseases such as musculoskeletal disorders, heart failure, diabetes mellitus, hypertension, dyslipidemia, or cancer, and could not undertake moderate exercise for any reason, and participants who already used the (Pacer)pedometer app.”

Please rephrase unclear language / correct grammar, for ex:

Following the screening visit, participants interested in the study were screened for pre-specified inclusion and exclusion criteria. Eligibility criteria included apparently healthy college students aged 18-30 years, who were? interested in a healthy lifestyle behavior , and with? ability to commit the necessary time and effort to be? participants in the study, with? a body mass index (BMI): BMI≥ 18.5, capable of performing walking PA , and owner of a? Smartphone (i-phone or Android). However, the exclusion criteria were participants who had no Smartphone, were in another program for weight reduction, were pregnant?, lactated?, had bariatric or recent surgery, or had any of the diagnosed chronic diseases such as musculoskeletal disorders, heart failure, diabetes mellitus, hypertension, dyslipidemia, or cancer, and or? could not undertake moderate exercise for any reason, and participants who already used the (Pacer)pedometer app”.

2-3

93 - 97

“The Randomization process occurred using random blocks to ensure there were similar numbers of participants in each study group. Using the "Research Randomizer ORG" computer software program (www.randomizer.org/form.htm),…..”

Please correct language / avoid repetitions, for ex:

“The Randomization process was performed by occurred using random blocks to ensure there were similar numbers of participants in each study group. Using the "Research Randomizer ORG" computer software program (www.randomizer.org/form.htm)…..”

3

103

“…after considering the following assumption [9-14] most RCTs have used two-sided…..”

A word is missing?

4

130 -133

“The first face to face (FTF) meeting (45-minutes) withall study participants was the introduction to the project. In this meeting, each groups consisted of 5 participants received a presentation about a healthy lifestyle, and the clinical benefits of PA for the recommended weekly duration.”

Please rephrase / correct grammar

4

141 - 145

“Commercially Available App. We used one of the most popular publicly available Smartphone apps for improvingthe PA (Step counts) (Pacer). It has goal setting functionality, self-monitoring of step

counts, calories expended, and automatic performance feedback through graphic display of step-count history”

Is the app validated?

4

154 - 156

“The first face to face (FTF) meeting (45-minutes) withall study participants was the introduction to the project. In this meeting, each groups consisted of 5 participants received a presentation about a healthy lifestyle, and the clinical benefits of PA for the recommended weekly duration.”

Please rephrase / correct grammar

4

162

“….phase (Baseline Assessment) until week 11, until they contacted for week 12 follow-up”

Please rephrase / correct grammar

5

171

“Objectively measured height, weight, BMI, and body fat percentage was obtained at the end of 12-week of intervention. ….”

No measured weight before intervention?

7

235 - 236

“…The use of pedometer based an m-Health app was found to increase the PA over a 12-week period, when compared with data from the control group. The intervention group achieved a significant increase of over 14575 steps per week (an increase of approximately 36 % in activity levels), ….”

Please comment on the validity of the app.

Author Response

1

12

……”m-Health”.

Please write the full wording and the abbreviation in a parenthesis the first time an abbreviation is mentioned.

The full wording Physical activity (PA) is written and highlighted.

1

27

….” Mean = 54896.”

Please write “steps per week” (if that is correct) after the step-counts. And please add in all step counts throughout the article.

Steps per week are written and highlighted.

2

46-47

“Mobile health….”

Please  full wording only first time mentioned.

Edited and highlighted.

55 - 58

….” With several studies highlighted an m-Health app intervention as a main channel in which included interventions that demonstrated its effectiveness in health-related changes, there is little attention to measuring the effect of an m-Health app intervention program designed to improve PA of college students..”

Please rephrase /correct grammar.

 Hard to understand the meaning.

The grammar is corrected and highlighted.

2

59-62

“Despite the potential mobile apps and growing interest in their utilization among the public and researchers, to the best of our knowledge, relatively few randomized control trials (RCTs) of m-Health apps as a healthy lifestyle intervention in itself that focuses on education and self-monitoring of PA. Thus, no such interventions have involved populations such as healthy college students [11- 12]. Therefore, the purpose of this a 12 week RCT is….”

Please correct grammar / add verb / rephrase

The grammar is corrected and highlighted

2

78-88

….” Following the screening visit, participants interested in the study were screened for pre-specified inclusion and exclusion criteria. Eligibility criteria included apparently healthy college students aged 18-30 years, interested in a healthy lifestyle behavior , and have ability to commit the necessary time and effort to participants in the study,

Please rephrase unclear language / correct grammar, for ex:

The grammar is corrected and highlighted

2-3

93 - 97

“The Randomization process occurred using random blocks to ensure there were similar numbers of participants in each study group. Using the "Research Randomizer ORG" computer software program (www.randomizer.org/form.htm),…..”

Please correct language / avoid repetitions, for ex:

“The Randomization process was performed by occurred using random blocks to ensure there were similar numbers of participants in each study group. Using the "Research Randomizer ORG" computer software program (www.randomizer.org/form.htm)

The language is corrected and highlighted

103

“…after considering the following assumption [9-14] most RCTs have used two-sided…..”

A word is missing?

The sentence is edited and highlighted

4

130 -133

“The first face to face (FTF) meeting (45-minutes) withall study participants was the introduction to the project. In this meeting, each groups consisted of 5 participants received a presentation about a healthy lifestyle, and the clinical benefits of PA for the recommended weekly duration.”

Please rephrase / correct grammar

The grammar is corrected and highlighted

4

141 - 145

“Commercially Available App. We used one of the most popular publicly available Smartphone apps for improvingthe PA (Step counts) (Pacer). It has goal setting functionality, self-monitoring of step

counts, calories expended, and automatic performance feedback through graphic display of step-count history”

Is the app validated?

The application is validated through a 20–step test which was carried out with each participant to calibrate the sensitivity level of the telephone for different individuals to ensure their step counts were recorded accurately.

4

154 - 156

“The first face to face (FTF) meeting (45-minutes) withall study participants was the introduction to the project. In this meeting, each groups consisted of 5 participants received a presentation about a healthy lifestyle, and the clinical benefits of PA for the recommended weekly duration.”

Please rephrase / correct grammar

The grammar is corrected and highlighted

4

162

“….phase (Baseline Assessment) until week 11, until they contacted for week 12 follow-up”

Please rephrase / correct grammar

The grammar is corrected and highlighted

5

171

“Objectively measured height, weight, BMI, and body fat percentage was obtained at the end of 12-week of intervention. ….”

No measured weight before intervention?

The body weight was measured before the intervention and mentioned in page 4, lines (121,122,123) “2.3.1. Anthropometric Measurements “At baseline (week prior to the intervention), anthropometric measurements included: weight, height, BMI, and percentage of body fat were obtained. Weight, BMI, and percentage body fat were determined using the TANITA Body Composition Analyzer (SC331S)”.

7

235 - 236

“…The use of pedometer based an m-Health app was found to increase the PA over a 12-week period, when compared with data from the control group. The intervention group achieved a significant increase of over 14575 steps per week (an increase of approximately 36 % in activity levels), ….”

Please comment on the validity of the app.

The validity of application was assessed through A 20–step test. We added this clarification to the Manuscript A 20–step test was carried out with each participant to calibrate the sensitivity level of the telephone for different individuals to ensure their step counts were recorded accurately. Page 4 line (135,136,137) .

Reviewer 3 Report

The first author has published a similar paper in the journal of "Current developments in nutrition" in 2019 with the same title, method, and result: 

"An m-Health Intervention Using a Smartphone App to Improve Physical Activity in College Students: A Randomized Controlled Trial (P16-025-19)"

Objectives

Determine the efficacy of a 12-week mobile health (m-Health) intervention with the goal of increasing daily step counts on physical activity, improve body mass index (BMI), and body fat mass among college students.

Methods

A 12- week randomized control trial was conducted. College students (n = 130) between 18–30 years of age were randomized to one of two conditions: Intervention (n = 65) and control (n = 65). All participants then had the Smartphone app downloaded onto their mobile phone to record their daily step count in order to provide a measurement of their baseline physical activity levels. Intervention group received physical activity goals of (10,000 step/day), information on the benefits of exercise, and automatic feedback. Control group received information on the benefits of exercise without any kind of intervention. The primary change was daily step count between baseline and follow-up.

Results

In this study, there were no significant intervention effects for BMI, fat mass and % body fat. Significant intervention effects were found for body weight (mean ± SE: 0.419 ± 0.164; P = 0.013). Physical activity as expressed by step counts significantly increased from baseline to post intervention (10,022 weekly/step; P = 0.008). Despite this, post intervention changes in outcomes were not significantly different from controls.

Author Response

We agree with the reviewer’s assessment regarding to this point. However it is just an abstract in American Society of Nutrition Annual Meeting /2019.

Round 2

Reviewer 3 Report

Accepted